# Evolution shapes and conserves genomic signatures in viruses
Martin Holmudden[1,4], Joel Gustafsson[1,4], Yann J. K. Bertrand[2], Alexander Schliep [3] &
Peter Norberg [1] ✉

The genomic signature of an organism captures the characteristics of repeated oligonucleotide patterns in its genome [1], such as oligomer frequencies, GC content, and differences in codon usage. Viruses, however, are obligate intracellular parasites that are dependent on their host cells for replication, and information about genomic signatures in viruses has hitherto been sparse. Here, we investigate the presence and specificity of genomic signatures in 2,768 eukaryotic viral species from 105 viral families, aiming to illuminate dependencies and selective pressures in viral genome evolution. We demonstrate that most viruses have highly specific genomic signatures that often also differ significantly between species within the same family. The species-specificity is most prominent among dsDNA viruses and viruses with large genomes. We also reveal consistent dissimilarities between viral genomic signatures and those of their host cells, although some viruses present slight similarities, which may be explained by genetic adaptation to their native hosts. Our results suggest that significant evolutionary selection pressures act upon viral genomes to shape and preserve their genomic signatures, which may have implications for the field of synthetic biology in the construction of live attenuated vaccines and viral vectors.

Genomes of organisms are shaped by the evolutionary interplay of selective forces and genetic drift. In addition to mutations leading to structural and functional changes of encoded proteins, that may be selected for or against, selection pressures also act on characteristics of the genome itself. More specifically, it has been demonstrated that many organisms have imprints of unique patterns in the arrangement and distribution of oligonucleotides within their genomes, referred to as the genomic signature[1]. These signatures typically remain conserved within species, reflecting the diverse selection pressures influencing their evolution. However, these selection pressures usually differ between species because of differences in intracellular environments and the cell-specific genetic machinery used, e.g., for maintaining, replicating, and reading the genome. Therefore, the genomic signatures typically differ between organisms. While initial studies on genomic signatures were often based on differences in GC content, this does not fully capture the specificity[2]. It has further been demonstrated that the genomic signature in a specific organism is visible in oligomer frequencies, nucleotide dependencies[3], differences in codon usage, and codon-pair bias[4]. One approach for analyzing genomic signatures is, therefore, to examine the frequency of oligonucleotides, so-called $k$-mers, which has been used to demonstrate that the genomic characteristics differ between animals[5],

different types of bacteria[6], and plasmids[7]. Based on the differences of genomic signatures among species, analysis of these is now widely used in a wide array of applications as an alignment-free method in comparative biology, without the need for traditional sequence alignment, as reviewed in refs. 8,9.

In contrast to cellular organisms, viruses are small infectious, intracellular parasites that infect all living organisms. Furthermore, viruses cannot replicate by themselves, and they thus utilize parts of the host cell's genetic machinery for replication and expression of their genetic material. Based on presumed selection pressures imposed by the host cell's intracellular environment and genetic machinery, viruses have been believed to mutate and adapt their genomes to increase viability in their respective host. Such host adaptation has, for example, been suggested for members of the *Flaviviridae* family[10] and for some bacteriophages that present similar GC content and codon usages to their hosts[11]. In addition, pairs of codons have been shown to deviate from their expected values based on individual codon frequencies assuming independence, possibly influencing the translation efficiency of genes[12]. By performing genome-scale changes in the codon pair bias in poliovirus, it was possible to attenuate the virus in mice while providing protective immunity[4]. Using a similar approach, it was possible to

[1]Department of Infectious Diseases, Section for Clinical Virology, Institute of Biomedicine, University of Gothenburg, Gothenburg, Sweden. [2]Laboratory of Molecular Biology and Bioinformatics, Institute of Botany, Czech Academy of Sciences, Prague, Czechia. [3]Department of Computer Science, Chalmers University of Technology, Gothenburg, Sweden. [4]These authors contributed equally: Martin Holmudden, Joel Gustafsson. ✉e-mail: peter.norberg@gu.se

attenuate a human respiratory syncytial virus[13] and the Influenza virus strain A/PR/8/34[14]. It was also demonstrated that replication of HIV in cell culture could be reduced by altering the codon-pair bias in the viral genome[15].

The underlying mechanisms of attenuation by de-optimizing the codon-pair bias have, however, been debated. Subsequent studies proposed that the attenuation observed after de-optimization of the codon-pair bias was instead a direct consequence of changes in the dinucleotide composition[16]. It was also demonstrated that the codon-pair and di-nucleotide biases are only marginally similar to that of their native host cells[16], suggesting that different selection pressures may act on viral and host genomes. Genomic signatures capture all these features while including several additional traits, which may allow for a more detailed analysis of genomes to identify differences in selection pressures acting on viral species, meaningful comparison of genomes, and identification of genomic viral-host adaptation.

The aim of the present study was therefore to analyze genomic signatures in the genomes of a large number of viral species to map the presence and specificity of their genomic signatures. We included the complete genome sequences of 2768 eukaryotic viruses from 105 different viral families in the analysis. Our results demonstrate that most viruses have highly specific genomic signatures. We further demonstrate that these signatures are conserved among the members of some viral families, while in other families the members present vastly distinct signatures. Most viral genomic signatures are also different from those of their respective hosts. We suggest that evolutionary selection pressures, primarily imposed by the viruses themselves, act on viral genomes to shape and preserve their genomic signatures.

## Results
### Genomic signatures in viral genomes

To investigate the degree of conservation of genomic signatures in viruses, we analyzed the complete genome sequences of 2768 viral species from 105 families. We applied established methods for the analysis of $k$-mer frequencies using variable-length Markov chains (VLMCs)[6,17,18]. These VLMCs are generalizations of models containing frequencies of fixed-length substrings commonly present in a certain genome, as explained in Fig. 1. Note that every such model counting substrings of fixed length is a VLMC. The converse is not true as VLMC not only adapts the depth of the tree representing the model to the statistics of the genome during training, but does so branch-by-branch. The motivation is to balance power—large k whenever frequent (k-1)-mer prefixes in a genome allow to reliably estimate probabilities of A, C, G, T following the same (k-1)-mer prefix—and robustness by only allowing large k if the counts for the prefix are sufficiently large. In practice, a maximal k is prescribed for VLMC. The final choice however, and whether that depth applies to all or some prefixes, is always automatically inferred from the genome sequence.

To avoid possible bias caused by repeat regions, all genomes were trimmed using DustMasker for removal of low complexity regions prior to any analysis. For viruses with segmented genomes, each segment was analyzed separately, totaling 4273 viral sequences. We divided each sequence into two parts: the first 30% termed *query* and the last 70% termed *profile*. Then, we compared the genomic signature in each *query* to the signatures in all *profiles*. If a query signature was most similar to its own profile signature, we considered it a conserved species-specific genomic signature, distinguishable from the signatures of all other viruses. For

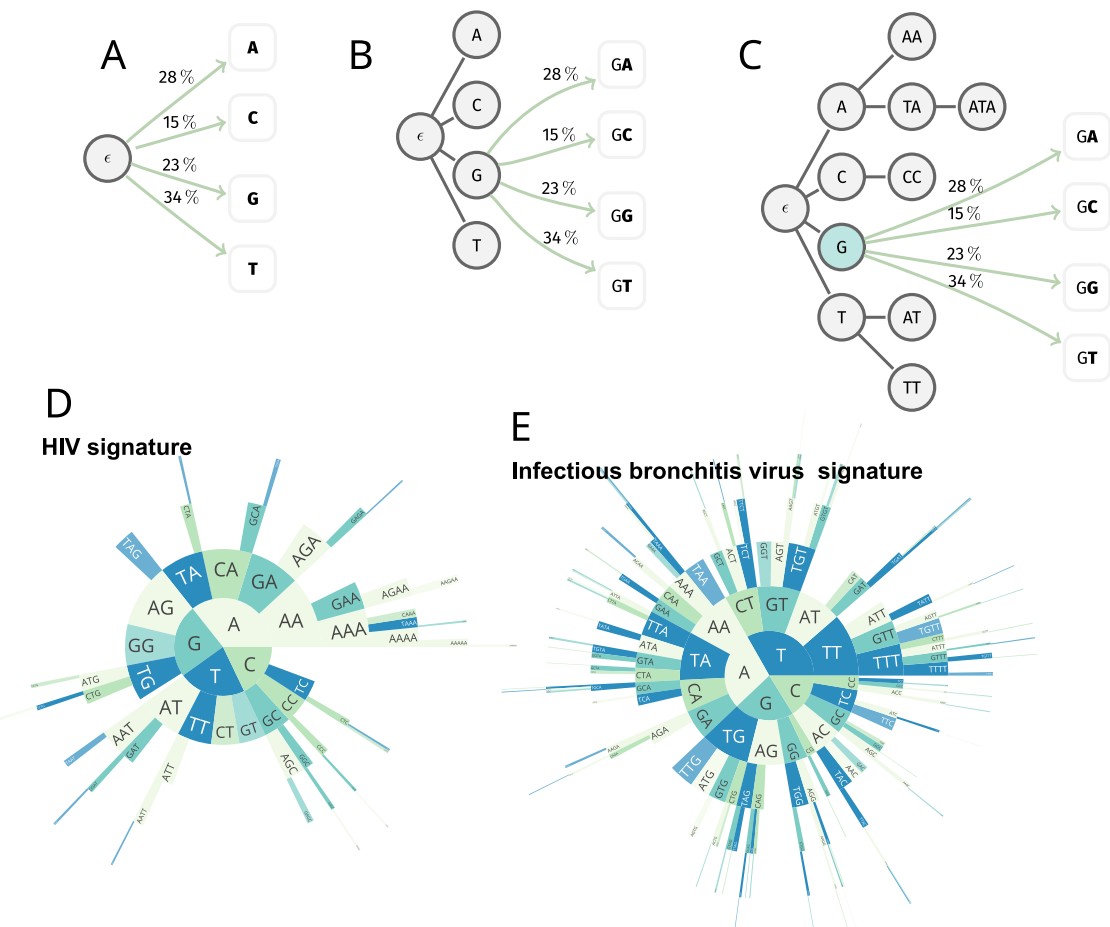

**Fig. 1 | Variable-length Markov chains (VLMC) are generalizations of models relying on frequencies of fixed-length substrings such as individual. A** di- or tri-nucleotides, codons, di-codons or, generally, k-mers. **B** Depicts a VLMC in which probabilities are assigned either to di-nucleotides starting with G or individual nucleotides not following a G. **C–E** Demonstrate the intrinsic balance learned during the training.

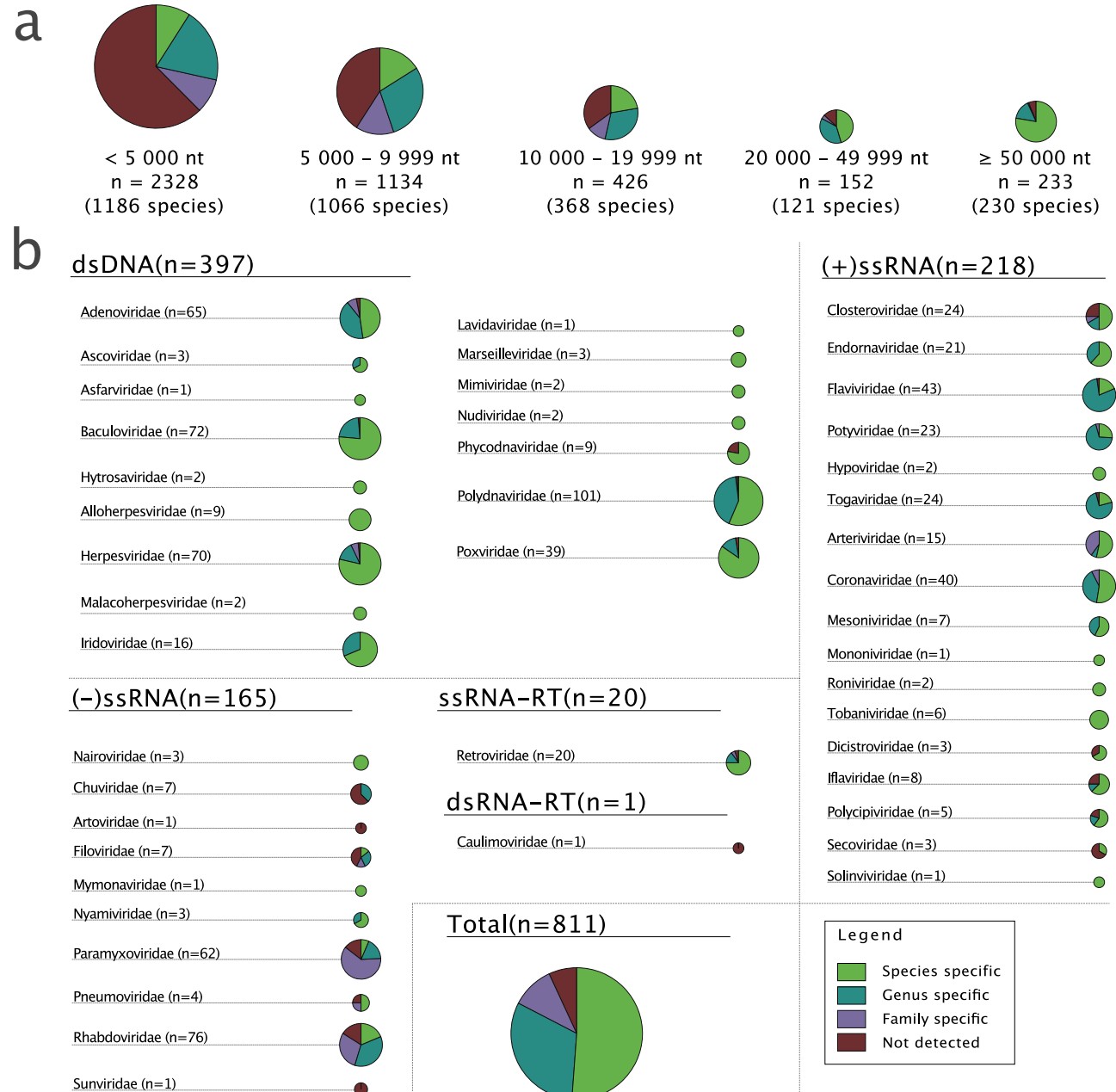

**Fig. 2 | Analysis of genomic signatures in viral genomes. a** Signature specificity was determined for viruses with genomes of different lengths. The proportional specificity is color-coded and represented in circle charts for respective genome length. **b** The signature specificity was further determined for viruses with genomes longer than 10,000 nt from different Baltimore classes and taxonomic families.

segmented viruses, a match to any segment profile of the same species was also considered species-specific. Although a certain virus has a conserved genomic signature, this signature might in some cases be highly similar to the signatures in other members of the same genus or family. Reasons for this could be either homologous genomes, and thus also similar signatures, or similar selection pressures acting on these viruses. The signature in the *query* sequence of that virus can therefore, in such cases, not be distinguished from the signature of the *profile* sequences of other viruses in the same genus or family. To highlight that these viruses present conserved genomic signatures, but that they are indistinguishable between other members in the same genus or family, we choose to call these signatures genus- or family-specific depending on the match. Consequently, if a *query* signature matched a *profile* signature of a different viral species in the same genus or family, we classified it as genus- or family-specific, respectively.

Our results show that viral genomic signatures are highly specific, often at the species level. To first explore the influence of genome size, we examined viral genomes of different sizes; ≤5000, 5000–9999, 10,000–19,999, 20,000–49,999, and ≥50,000 nucleotides (nt). Species-specificity was most prominent in viruses with large genomes, gradually decreasing with genome size (Fig. 2a). More specifically, 78% of all viruses with genomes longer or equal to 50,000 nt presented species-specific genomic signatures, distinct from other viruses from the same or different viral families, regardless of genome length. Among the remaining 22%, most had genus- (15%) or family-specific (1%) signatures. Only 6% of these didn't match any signature from the same family.

Viruses with genomes ranging from 20,000 to 49,999 nt presented similar results, although with fewer viruses with species- (45%) and more

with genus- or family-specific signatures (38% and 4%, respectively). In contrast, 13% of these viruses had genomes where the query did not match any profile signature from the same family.

For viruses with genomes between 10,000 and 19,999, and 5000–9999 nt, 22% and 16% had species-specific, 31% and 29% genus-specific, and 12% and 14% family-specific signatures. In these groups, 34% and 41% of the viruses presented genomes where the query did not match any signature from the same family.

Lastly, for viruses with genomes under 5000 nt, 9% presented species-specific, 19% genus-specific, and 9% family-specific genomic signatures. In contrast, for 62% of these viruses, we found no match between the query and any profile within the same family.

To evaluate the statistical robustness of our results, we applied a simulation approach where we randomly paired queries and profiles and compared the number of matches with our results using a Bonferroni-corrected two-tailed $t$-test. This test demonstrated significant results for all size groups ($p = 3.6 \times 10^{-12}$, $3.1 \times 10^{-15}$, $5.8 \times 10^{-13}$, $2.9 \times 10^{-13}$, $1.4 \times 10^{-14}$ for the family-specificity per category in order of increasing sequence length, with even lower $p$ values for the genus- and species-specific matches, Supplementary Fig. 1).

## Impact of sequence length

Our analysis of genomic signatures relies on repetitive nucleotide patterns, which implies that a higher specificity is typically achievable in longer sequences due to a higher prevalence of each repeated $k$-mer. As a consequence, the lower frequencies of detectable genomic signatures in viruses with shorter genomes presented here may be derived from a methodological bias.

We therefore tested if the lower detection of genomic signatures in shorter genomes could be attributed to a methodological bias, or to less prominent signatures. We randomly extracted subsequences of 5000, 10,000, and 20,000 nt from the largest genomes (>50,000 nt) in our dataset. We then analyzed these subsequence segments for species-, genus-, and family-specific genomic signatures and compared the results to our previous findings of similar subsequence length (Fig. 2a). To minimize the impact of subsequence location in the genome, this process was repeated 100 times, and the results were summarized.

With 5000 nt subsequences, species-specificity dropped from 78% to an average of 42% (Fig. 3a). Still, it's notably higher than the 22% seen in viruses with genomes between 5000 nt and 9999 nt. Likewise, genus- and family-specificity decreased from 91% and 94% to 53% and 66% (Fig. 3b, c), but still more accurate than viruses of similar length. This trend held for 10,000 nt and 20,000 nt subsequences, affirming the correlation between larger genomes and increased genomic signature specificity in viruses.

## Impact of k-mer length

Here, we use a variable length Markov model to analyze genomic signatures where we state a maximum k-mer length. Previous studies on procaryotes have indicated that the predictive accuracy increases with k-mer size. However, the increase appears to be logarithmic, where an increased size from two to four nucleotides significantly increases the accuracy, while an increase from five to six, or six to eight nucleotides only marginally improved the results[6,19]. A similar study demonstrated that the accuracy of classification was good already using k-mers of three nucleotides, while longer k-mers of five nucleotides improved the results[20]. Similarly, in a study using genomic signatures for phylogenetic studies, the authors found an improvement with a k-mer length between two and five nucleotides, while it remained stable with increased length. It was subsequently concluded that a k-mer of length six nucleotides presented a good trade-off between sequence size and k-mer length, which were chosen for further studies[21].

To investigate to which extent the maximum k-mer length affects our results on viral genomes, we repeated the analysis of all viral genomes using maximum k-mer lengths of one to seven nucleotides. Our results show that the optimal max length is not necessarily as large as possible but varies depending on genome length. While the genomic signature in viruses with short genomes was better analyzed with a max length of six or seven nucleotides, a shorter max length presented more accurate hits on species, genus, and family levels for viruses with larger genomes (Fig. 4). For simplicity and consistency, we decided to apply a maximum length of six nucleotides in all our analysis, which we consider a proper balance between computational time and accuracy.

## Families and Baltimore classes

To examine the variation in signature specificity among different viral families and Baltimore classes, we performed a new analysis on the 811 viral genome sequences exceeding 10,000 nt to avoid possible bias caused by analyzing short genomes.

We observed more than 50% species-specific and over 75% genus-specific genomic signatures for these viruses (Fig. 2b). However, differences existed among Baltimore classes and families. In Baltimore class I (dsDNA viruses), the majority, especially in *Baculoviridae* and *Herpesviridae*, presented species-specific genomic signatures (Fig. 2b). Baltimore class IV ((+) ssRNA viruses) generally displayed high specificity, although slightly lower at the species level. For instance, in *Coronaviridae*, 53% presented species-, 40% genus-, and 7% family-specific genomic signatures. The lowest specificity was found in Baltimore class V ((-)ssRNA viruses). For example, *Paramyxoviridae* and *Rhabdoviridae*, the two largest families, only presented 6% and 19% species-specific genomic signatures, respectively. Nevertheless, most (-)ssRNA viruses, including those in *Paramyxoviridae* and *Rhabdoviridae*, exhibited at least family-level specificity.

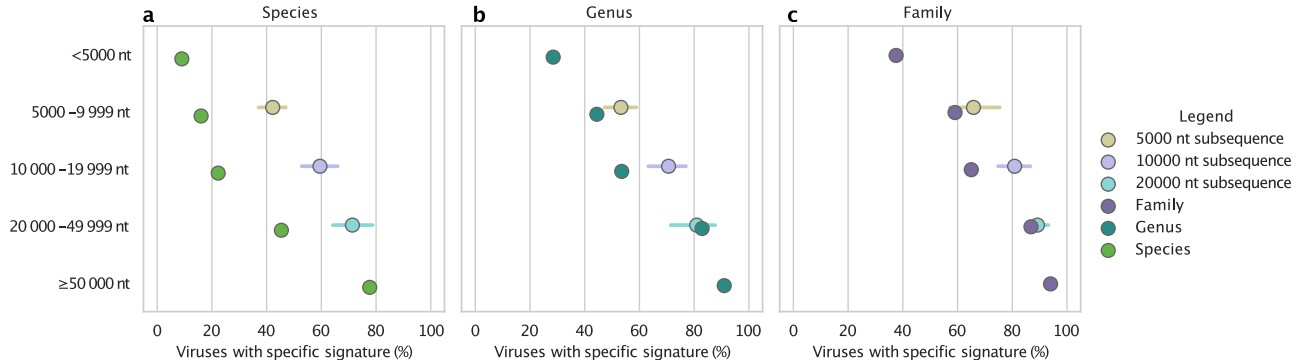

**Fig. 3 | Evaluation of methodological bias related to sequence length.** The genomic signatures of subsequences from viruses with genomes longer than 50,000 nt were compared to those of viruses with genomes of the corresponding sequence lengths. The analysis demonstrates that the (**a**) species-, (**b**) genus-, and (**c**) family-specificity for subsequences of 5000 nt (yellow), 10,000 nt (light purple), and 20,000 nt (light blue) from the sampled viruses have, on average, a higher fraction of specific signatures. Our results suggest that larger genomes tend to have more significant genomic signatures than smaller genomes, although there is also a methodological bias. The 95% confidence intervals are depicted as horizontal bars.

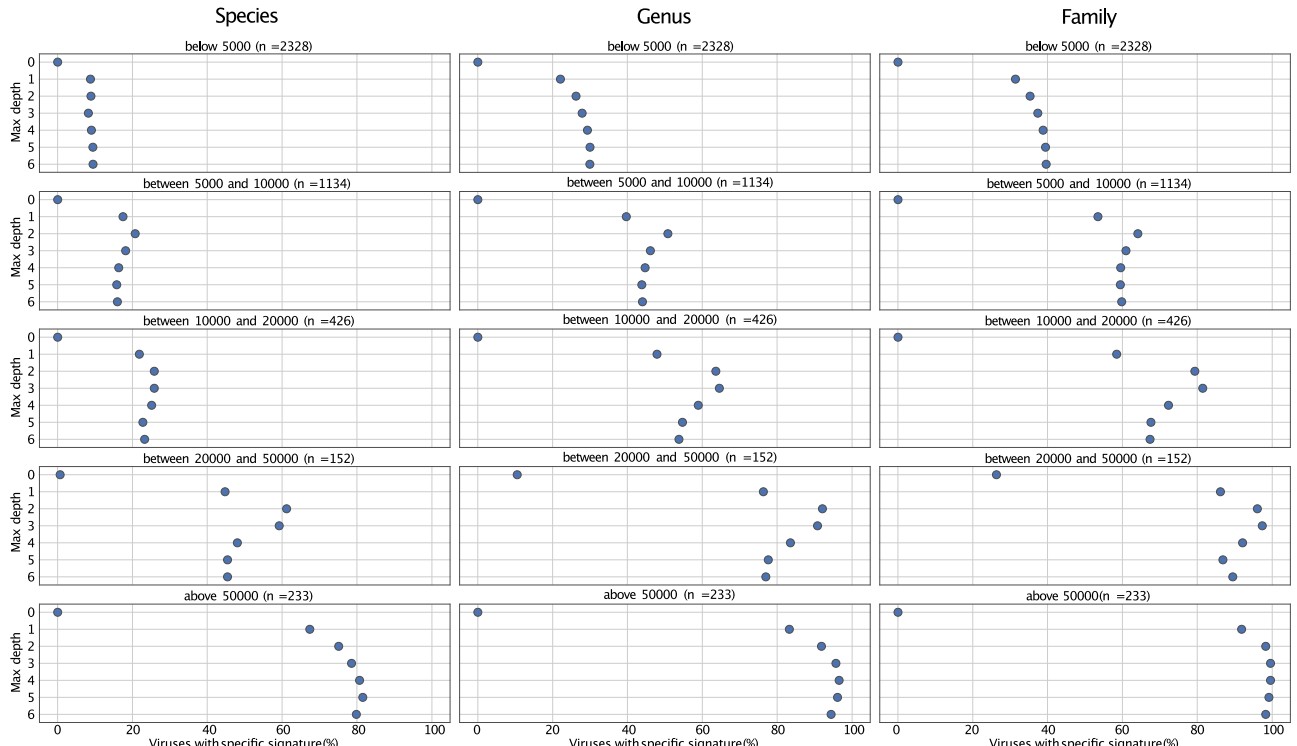

**Fig. 4 | Impact of k-mer max length.** The genomic signatures of viral genomes of different lengths were analyzed using seven different k-mer max lengths (y-axis). Our results demonstrate that the optimal k-mer max length differ for viruses with different genome sizes.

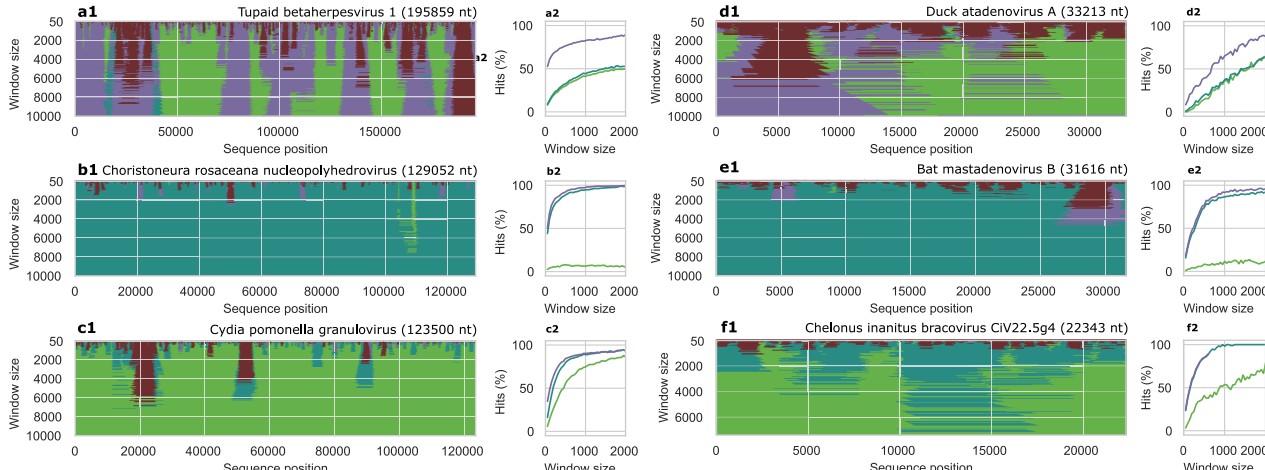

**Fig. 5 | Genomic signatures across genomes. a1–f1** We randomly selected six viruses that we had classified as containing family-, genus-, or species-specific genomic signatures, respectively, and applied a sliding window approach to analyze the species- (light green), genus- (dark green), or family-specific (purple) signatures in different regions in their genomes. Regions with no specificity related to the viral family is marked in red. **a2–f2** We further depicted the proportion of windows that is most similar to the correct species, genus, and family for respective virus to estimate the possibility to classify a viral sequence based on its genomic signature for different sequence lengths. The viruses for each panel correspond to the virus with the same letter as in (**a1–f1**).

A limited number of Retroviridae family members exceed 10,000 nt and were included from the ssRNA-RT viruses (Baltimore class VI). Among them, 75% showed species-specific, 15% genus-specific, and 5% family-specific genomic signatures. Only 5% of the viruses from this Baltimore class did not present a detectable genomic signature.

In dsDNA-RT (Baltimore class VII), only one virus exceeds 10,000 nt —the *Cacao swollen shoot virus* in the *Caulimoviridae* family, which had no discernible species-specific genomic signature.

We assessed the statistical significance of species, genus, and family-specific signatures for Baltimore classes, using the same statistical test as for the size groups (Supplementary Table 1). Except for

*Solemoviridae* ($p = 0.18$), *Nanoviridae* ($p = 0.43$), and *Marnaviridae* ($p = 1$), all viral families demonstrated statistically significant fractions of their members with either species-, genus-, or family-specific signatures as compared to the random model ($p < 0.05$, Bonferroni corrected).

### Variation within genomes

To analyze intra-genomic variations in signature conservation in viral genomes longer than 10,000 nt, we applied a sliding window approach. The window size ranged from 50 nt to 10,000 nt, or at most one-third of the genome length.

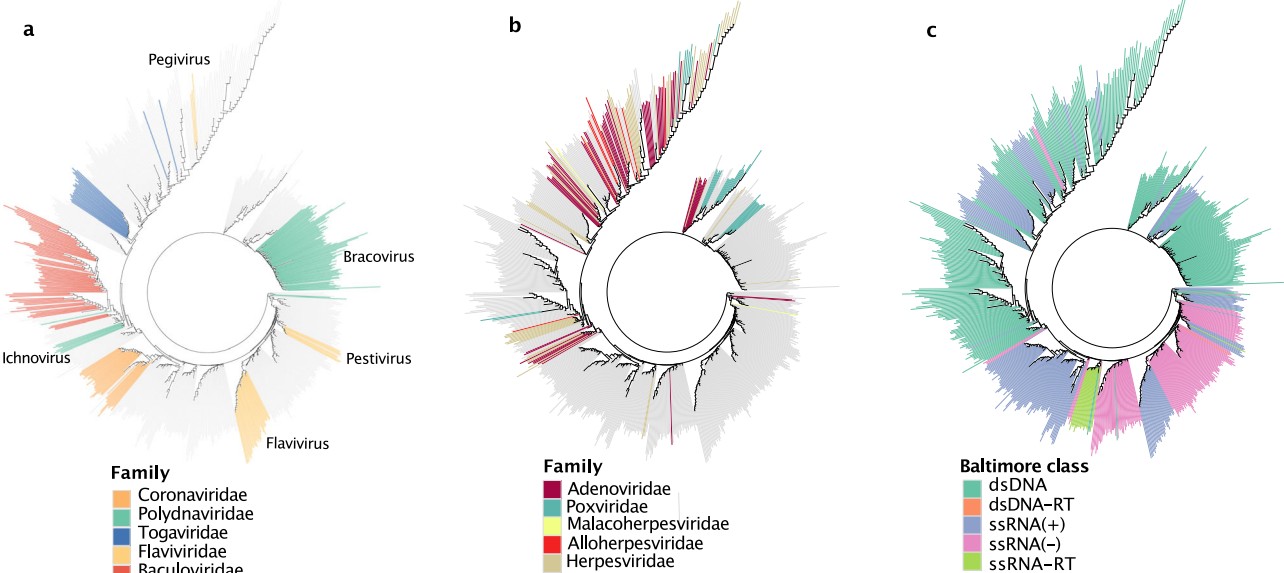

**Fig. 6 | Clustering of viruses based on their genomic signatures.** We constructed an unrooted tree of the viruses with genomes larger than 10,000 nt, where the distances between taxa correspond to differences in genomic signatures. **a** A subset of the families, where the species of the same genera have similar signatures. **b** A subset of the families, where signatures varied significantly within each family. **c** The respective Baltimore class is color-coded to illustrate their respective variation in genomic signatures.

We analyzed six viral genomes: three with sequence lengths between 20,000 nt and 49,999 nt, and three exceeding 50,000 nt. Species were randomly selected from viruses with species, genus, and family-specific signatures within their respective length group.

Most signatures were conserved across the analyzed sequences (Fig. 4a1–f1). For *Cydia pomonella granulovirus* (Fig. 4c1) and a segment from the *Chelonus inanitus bracovirus* (segment proviral CiV22.5g4 gene) (Fig. 4f1) species-specificity spanned nearly their entire sequences, with only a few non-matching regions, mainly repeat regions.

In *Choristoneura rosaceana nucleopolyhedrovirus* (Fig. 4b1) and *Bat mastadenovirus B* (Fig. 4e1), most regions were genus-specific. *Duck atadenovirus A* (Fig. 4d1) has two distinct regions: one mostly family-specific and the other mostly species-specific. *Tupaiid betaherpesvirus 1* (Fig. 4a1) displays regions with predominantly species- or family-specific windows. For all viruses, windows shorter than 200 nt typically failed to match even the correct family.

We also explored the minimum sequence length needed to identify the species, genus, or family by analyzing genomic signatures (Fig. 5a2–f2, Supplementary Fig. 2). On average, analyzing 1000 nt can classify the correct family in 80% of viruses with genomes longer than 10,000 and the correct genus for 70% (Supplementary Fig. 2). Identifying the correct species is more challenging; analyzing 2000 nt can classify less than 40% of species, although results vary by species. For example, *Cydia pomonella granulovirus* (Fig. 4c2) requires just 500 nt to identify the correct species in 50% of cases, while *Bat mastadenovirus B* (Fig. 4b2) can achieve accurate genus classification at best.

## Variations within and between families

To further explore how genomic signatures differ between species within and between families, we first computed pairwise signature distances between all viruses with genomes larger than 10,000 nt. These were then used to create an unrooted neighbor-joining tree. This tree thus illustrates similarities and differences in genomic signatures between viruses based on their locations, rather than descendance from common ancestors like in a phylogenetic tree.

Our results demonstrate that some viruses clustered according to their families or genera, such as *Corona, Polydna, Toga, Flavi*, and *Baculo* families. However, while all viruses within individual *Flaviviridae* genera (*Flavivirus*,

*Pestivirus, Pegivirus*) presented similar genomic signatures, there were large distances between the genera (Fig. 6a). Similarly, *Polydnaviridae* presented conserved signatures within, but not between, genera (*Bracovirus, Ichnovirus*).

While all viruses in the *Baculo* and *Corona* families, and most viruses in the *Toga* family, clustered according to their family taxonomy, the *Baculoviridae* branch also included 15 distantly related, or unrelated, viruses. Additionally, three *Tobani* family viruses had similar signatures to, and clustered together with, viruses in the *Corona* family. In contrast, in some viral families, such as the *Herpesviridae, Alloherpesviridae, Malacoherpesviridae, Adenoviridae*, and *Poxviridae*, the genomic signatures varied considerably within the families and the members were dispersed throughout the tree (Fig. 6b).

Although some viral families within the same Baltimore class presented similar genomic signatures (Fig. 6c), multiple clusters existed of each Baltimore class, except for the ssRNA-RT viruses. It's important to note, however, that the *Retroviridae* family is the sole family within this class.

A high-resolution tree with detailed information is presented in Supplementary Fig. 3.

## Viral-host signature adaptation

As viruses partly depend on their host's genetic machinery, they may have adapted their genomic signatures to converge with their hosts' signatures. We tested this hypothesis using 2399 host genome sequences, including putative hosts, vectors, and reservoir hosts. We computed genomic signature profiles on host coding regions and compared them to viral signatures. Since many related eukaryotic species share sequence homology and thus likely have similar signatures, we counted matches on hosts from the same genus, family, and order as a positive match.

We found that only 45 viruses presented genomic signatures that were most similar to those of a host of the correct species, genus, family, or order. Six viruses had signatures similar to the signature of the correct host species, namely one *Retroviridae* (*Murine leukemia virus*, which includes endogenous subspecies), three members of *Potyviridae*, one member of *Iridoviridae*, and one *Hepadnaviridae* (Supplementary Table 2). Expanding the criteria to host order revealed more matches than expected at random (Fig. 7). For three Baltimore classes, some viral families presented significant host-order similarity: dsDNA (*Hepadnaviridae, Nudiviridae, Ascoviridae,*

**Fig. 7 | Similarity analysis of genomic signatures in viruses and their hosts.** The percentage of viruses similar to a host within the same taxonomic order as its native host (in orange), compared to a random model (in gray). The viruses are subdivided by their genome composition. Viral families with significantly more viruses similar to their hosts than randomly expected are marked (*).

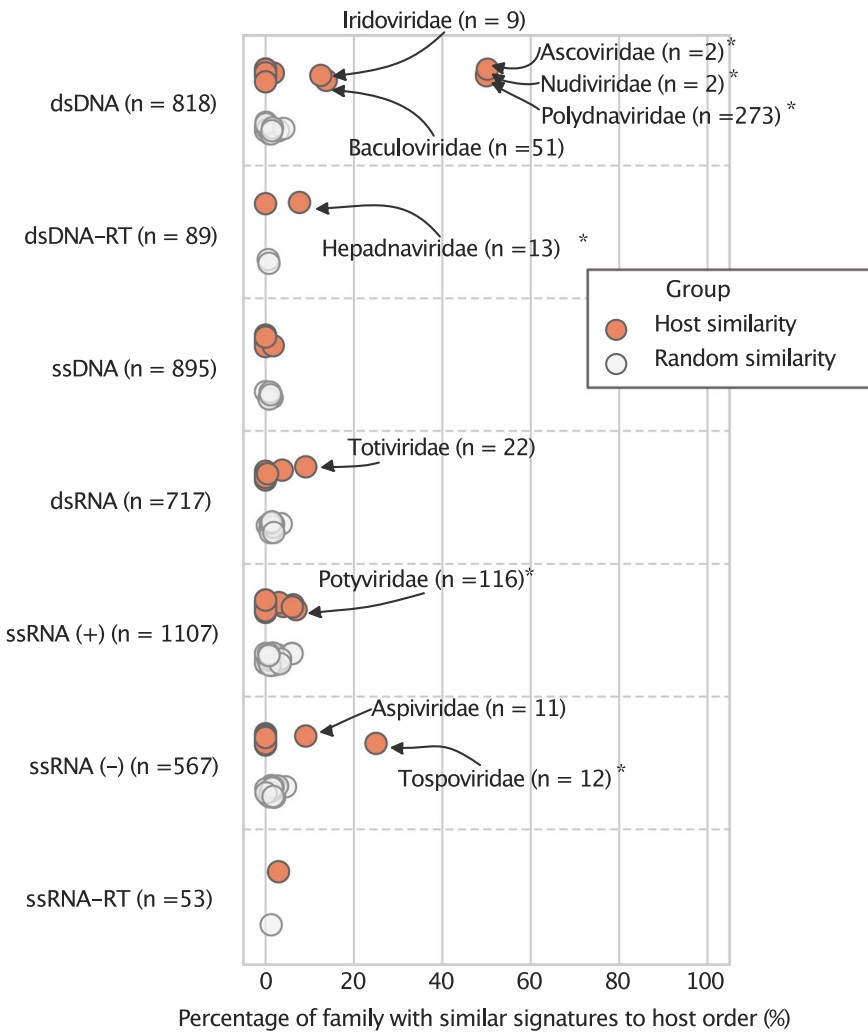

*Polydnaviridae, p < 3.6 × 10⁻¹⁰*), (+)ssRNA (*Mesoniviridae, Potyviridae, Betaflexiviridae p < 0.005*), (-)ssRNA (*Tospoviridae p = 0.005*). Plant and insect viruses were overrepresented among viruses with matching genomic signature to their hosts. Specifically, 19 viruses were plant viruses, 16 were insect viruses, one virus infects insects and mammals, two viruses infect both insects and plants, three infect other animals, and four fungi viruses.

## Discussion

### Species-specific genomic signatures

In this study, we demonstrate that most viral genomes contain genome-wide specific signatures, which are often distinct enough to allow for viral species discrimination. We also show that viruses with large genomes generally have more specific signatures than viruses with shorter genomes, and that the signatures are often preserved across the genomes. In addition, the different species in some families, such as the *Coronaviridae*, present similar genomic signatures, while there are significant variations in signatures between the members in other families, such as the *Herpesviridae*. Finally, although viruses in varying degrees are dependent on their hosts' cellular genetic machinery and environment, viral genomic signatures are typically distinct from those of their respective target host cells. We propose that viral genomic signatures, at least partly, result from various selection pressures acting on viral genomes. The dissimilarity to the signatures of their respective hosts suggests that there are different selection pressures acting on viral and host-cellular genomic signatures.

Although the specific biological mechanisms responsible for those selection pressures are currently not fully understood, our results suggest that they differ between viral species and families. A deeper knowledge about the details and differences of genomic signatures of various viral species and families may reveal vital information about specific genetic mechanisms and selection pressures, and this research field may thus present an interesting avenue for further in-depth studies of genomic signatures in viruses of certain interest. Nevertheless, various general mechanisms have been proposed to influence genomic signatures, that may also be applicable for viruses. For example, the GC content of a genome is related to thermal stability[22], thus making different GC content preferable in different environments. A specific GC content invariably results in a particular set of k-mers, which influences the genomic signature and is likely one contributing factor to the signatures in viruses observed here. It is also well known that many cellular genomes have a preferred set of codons for which their gene expression machinery is optimized[23,24], although some cellular genomes present a significant degree of variation in codon usage[25]. Another example codon bias and codon pair bias, which are factors that influence translation efficiency, and folding and stability by direct effect on secondary structure of viral RNA. Viruses have also been shown to adapt to and be mutated by the host's immune defense, for example, by the protein families APOBEC[26], ZAP[27], and ADAR[28,29]. In the long-term, changes caused by avoiding the immune system or induced by the immune system may lead to an imbalance of the frequency of k-mers throughout a genome and thus influence the genomic signature. As mentioned in the introduction, however, many parts of a genomic signature are dependent on each other. The codon-pair bias cannot be changed without altering the di-nucleotide motifs, and neither can the codon usage be changed without altering the GC content, etc.

Although we noticed that the GC content seemed to be a strong contributor to the genomic signatures, we also treated all k-mers equally, without relation to reading frames and position in the genomes, and hence did not distinguish between codons and "off-frame" triplets of nucleotides. We therefore made no further attempts to highlight the different contributors to the genomic signatures in different viral species in the present study, although it would be an interesting topic for future studies.

We found that less than 50% of the viruses with short genomes (<5000 nt) presented specific genomic signatures. While we demonstrated that this is partially a methodological bias, there may be other biological factors that lower the signature specificity. For instance, smaller viral genomes do not carry the same number of auxiliary genes that can help with genome replication and proofreading, which may influence the genomic signature. Additionally, in short sequences, every random mutation in the genome has a proportionally more prominent influence on the signature, as any mutation changes a larger proportion of the repeated k-mers.

In addition to various selection pressures shaping and conserving viral genomic signatures, we cannot exclude that also non-beneficial mechanisms contribute. For example, in the *Hepatitis C virus*, the replication machinery typically introduces a specific set of errors throughout the genome[30], and such systematic mutations may partly modify the viral genomic signature.

## Conservation across genomes

We demonstrate that the genomic signatures are typically genome-wide conserved, which suggests that there are similar mechanisms acting throughout genomes. There are, however, some distinct regional variations in the signature discriminatory power, which are mostly related to repeat regions. Repeat regions are sequences present in multiple adjacent copies in genomes, which, by definition, give rise to repeated oligonucleotides that will alter the genomic signature. Nevertheless, these genome-wide conserved viral genomic signatures present a possibility of classification of short viral fragments since it was in many cases possible to classify a sequence of only 500 nt to the correct family and sometimes even the correct species (Fig. 5).

## Variations within and between families

We observed that there is a large variation in the degree of conservation of genomic signatures within viral families and genera. Some families have a prominent family-wide genomic signature, some genera have genus-wide signatures, and a few smaller subgroups displayed similar but distinct signatures.

If the genomic signature is highly similar for several members, it is difficult to distinguish between them. This close similarity between some viruses' signatures may partly explain why we observed a relatively small degree of species-specific signatures in Fig. 2b for some families, even if they presented genome-wide conserved signatures that were similar to the signatures of other viruses in the same genus or family.

There may be several underlying causes for why some viral families present conserved signatures among their different species, while others do not. A simple explanation may be that some families present more diverged genomes than others, and therefore also more diverged genomic signatures. After comparing the sequence divergence of the members of the families in Fig. 6a, b, we indeed noted that the families with more conserved genomic signatures typically also presented higher sequence identities. This is likely also the reason for the conservation of genomic signatures in the *Corinaviridae* family, that was suggested to have diverged over 150 million years ago[31], but which genomes likely have been conserved through purifying selection[32]. We found, however, some viruses that presented similar genomic signatures while being more distantly related. One example is the three viruses from the *Tobaniviridae* family (porcine- and bovine torovirus, and bovine nidovirus) that clustered together with *Coronaviridae* (Supplementary Fig. 3), although being phylogenetically distinct from those. These belong to the *Nidovirales* order, and despite the close similarity in genomic signatures with *Coronaviridae*, other members of *Nidovirales*, including

more closely phylogenetically related viruses, such as other members of *Tobaniviridae*[33], presented widely different genomic signatures. Another example is the family *Baculoviridae*, of which members presented similar genomic signatures, and to which also 15 other viruses clustered. Of these, three belong to the same order (*Lefavirales*) as *Baculoviridae* (two nudiviruses and one hytrosavirus), while the other eleven are unrelated or more distantly related to *Baculoviridae* (three iridoviruses, three ascoviruses, two poxviruses, three phycodnaviruses, and one polydnavirus). While the underlying reasons for this similarity in genomic signatures of unrelated, or distantly phylogenetically related, viruses are unknown, it is likely that common selection pressures act on these viruses. Although these selection pressures may be elucidated in detail in future studies, we noticed that some viruses presenting similar genomic signatures also infect similar host species. For example, the three tobaniviruses that presented similar signatures as the *Coronaviridae* family also infect mammalian hosts, as coronaviruses do, while the remaining tobaniviruses that presented dissimilar genomic signatures have fish- or reptilian hosts and cluster with two *Roniviridae* viruses (Supplementary Fig. 3). All three families belong to the *Nidovirales* order. In addition, baculoviruses infect insects, and among the 15 other viruses presenting similar genomic signatures, 9 also infect insects. Another observation was that for the *Flavivirus* genus, tick-borne viruses grouped separately from the mosquito-borne and no-known-vector viruses. We can therefore not exclude the possibility that the host cellular environment may induce various selection pressures on viral genomes, and that viral host adaptation may contribute to shaping the genomic signatures of some viruses[10]. It has also been suggested that bacteriophage evolution differs by host and host range[34], and an intriguing question is whether a broad host range may infer restrictions on the specificity of the genomic signature in a specific viral species. We noticed for example that the extremely narrow host range viruses in the *Herpesvirales* order not only presented species-specific genomic signatures, but these were also vastly different between the members of this group, illustrated by their positions in the neighbor-joining tree. A similar pattern was, however, also seen for members of the *Adenoviridae* family, which are known to have a much broader host range. In conclusion, although the genomic signatures of some broad host range viral families, such as the *Coronaviridae* family, presented conserved genomic signatures within the family, we could not find an unequivocally convincing correlation between host range and genomic signatures in viral genomes.

## Host adaption

Among the viruses that presented similar genomic signatures to those of their hosts, members of the *Polydnaviridae* family were most prominent. However, this group of viruses is endogenous following likely multiple insertions of the genome in a precursor to the modern hosts[35]. Therefore, the detected host similarity may stem from the incorporated viral sequences in the host genomes, rather than from similar genomic signatures. To remove this possible source of bias, we excluded all endogenous viral sequences from the host sequences and repeated our analysis, which did not change in results. A possible explanation of the viral-host similarities in genomic signatures may thus be that by being integrated and expressed by the host's cellular machinery, the genomic signature of these viral sequences has adapted to those of their host cells. We cannot, however, exclude alternative explanations. It has, for example, been proposed that polydnaviruses evolved partly through the acquirement of genes from wasps and other species through horizontal gene transfer (reviewed in ref. 36). It is therefore possible that the similarities in genomic signatures may, at least partly, be explained by incorporated host genes in the viral genome. Another example of host integration is the retrovirus *Murine leukemia virus*, that also presented similar genomic signatures to its host.

We found that only a minority of viruses presented similar genomic signatures to those of their hosts. This may seem surprising, considering that many bacteriophages have been demonstrated to often present similar genomic signatures to those of their host cells[11].

Some studies have suggested that also many features of eucaryotic viral genomes, such as dinucleotides[10,16,27], codons[37], and codon pairs[4], may influence the viability of a virus in its host. On the other hand, previous studies have demonstrated that such similarities are rare and that most viruses typically present different codon usage than their host cells[38–41].

Nevertheless, even if a host adaption of viral genomic signatures would increase features such as the replication rate, it has been demonstrated that a higher degree of replication of the HIV genome decreases virulence because of an increased host immune system response[42]. Therefore, there may not be selective pressures to increase the replication rate beyond some threshold, which may partly explain why we do not observe host-similarities for most viruses.

We finally note that long-term viral-host co-evolution does not necessarily lead to an adaptation of the viral genomic signatures to those of their respective host cells. For instance, the *Herpesviridae* viruses have co-evolved with their hosts for hundreds of thousands, or millions of years[43], but often present dissimilar genomic signatures to their hosts. A striking example is the two alphaherpesviruses herpes simplex virus type 1 (HSV-1) and varicella-zoster virus (VZV) that both have humans as their only host, replicate in epithelial cells, and establish latency in sensory nerve ganglia. Despite this, these viruses present vastly different genomic signatures, not only to their human host, but also between each other (Supplementary Fig. 3). A possible explanation for this may be that they, despite their similar biology, have different niches in the human host and that the cellular environment differs in these. Another explanation may be that the selection pressures that form and preserve their genomic signatures are, at least partly, imposed by their own viral proteins that have diverged due to selection and/or genetic drift. Herpesviruses have, for example, been shown to harbor several genes that encode for proteins that control and manipulate the host cell environment. One such example is so-called host shutoff proteins that downregulate the host gene expression. While the encoding genes may differ between the members of the *Herpesviridae* family, they typically downregulate or inhibit the function of messenger RNA in the host cell, as reviewed in refs. 44,45. It could be speculated that the very long co-evolution with their hosts, and their many functions to regulate the host cell environment in different ways, have made herpesviruses partly independent on their host cell environment, which may to some extent explain the divergence of the species-specific genomic signatures among herpesviruses.

### Maximum k-mer size

While previous studies on bacterial genomes suggest that the predictive accuracy in analyzing genomic signatures increases with k-mer size[6,19], our results suggest that the optimal max size differs between viral species. A longer maximum k-mer size did thus not necessarily yield better results. Although this may partly be explained by random events, it is reasonable to assume that larger k-mers are generally less frequently present in a genome than smaller k-mers, such as codon or di-nucleotide motifs. Consequently, these will only be repeated to the extent to be included in the signature if the sequence under analysis is sufficiently long. A possible explanation to our results demonstrating that the results decreased with larger k-mers for many viruses may therefore be that although the profile includes the longer k-mers, these may not be present frequently enough in the query sequences since these are much shorter than the profile sequence. We can therefore not exclude the possibility that longer k-mers may have been selected for and may have some importance in some viral genomes, but that our methods are not sensitive enough to detect them in short sequences.

### Possible limitations

We have identified a few possible limitations with our study. First, repeat regions undoubtable infer bias due to changes in the relative frequency of any k-mer included in these regions, which is why these were removed using DustMasker prior to analysis. We can, however, not exclude the possibility that some short repeats may remain in some regions, which can infer some bias. Although such regions may still be present in some genomes, it is likely that they will only be found in either the query or the profile, leading to fewer correct matches rather than more. As such, even if some repeat regions may still be present in some genomes, the effect on our results is likely small, and the results would likely rather be slightly improved by removing such regions, should they exist.

Another limitation may be the possible bias inferred by duplicated genes. Since many viruses evolve partly through gene duplication, there is a possibility that in rare cases both the query and the profile sequences of a certain species contain copies of paralog genes that may bias the results. Our sliding window analysis showed, however, a large conservation across viral genomes indicating that duplicated genes should have little or no effect on our results.

In our virus-host analysis we analyzed complete genomes of viruses (excluding repeat regions), but only coding regions of the host genomes. The main reason for this was that the coding density in viruses is generally much higher and more complex with overlapping reading frames, etc, than in the host genomes and that the bias inferred by non-coding regions would likely be much higher in the host genomes. Using our sliding window approach, we also found that the genomic signatures in viral genomes are usually conserved, except for in repeat regions, and that the short non-coding regions did not drastically change the overall signature, which would limit this possible bias. Another possible bias in this analysis is the inclusion of endogenous viral sequences in the host genomes that, except for a few exceptions, were also included in the analysis. If these incorporated sequences have not yet mutated enough to adapt to the new genomes, inclusion of these may affect the overall signature leading to some bias. We therefore acknowledge that a future more in-depth analysis on viral-host similarities of genomic signatures may reveal additional information on viral-host adaptation and dependencies.

### Conclusions

We conclude that most viral genomes have conserved and distinct genomic signatures, and that these are likely shaped and preserved by various selection pressures. An implication hereof is that mutations that deviate from the genomic signature are less likely to be fixed in viruses from all Baltimore classes, including viruses with otherwise high mutation rates, such as retroviruses. We, therefore, suggest that genomic signatures may be used to improve predictions of evolutionary potentials of viruses[46] by improving the methods for quantifying and anticipating evolutionary events, as future substitutions are likely constrained by the requirements to preserve the genome's signature.

Genomic signatures may also be included in viral evolutionary studies by designing new evolutionary models that account for the variation of natural selection favoring certain genomic signatures across viral phylogenetic lineages. With a detailed knowledge about differences in signatures in different clades, these models could be tailored to clades individually, and therefore produce more accurate phylogenies and improve the dating of divergence events.

Other applications may be to use viral genomic signatures for the detection of recombination and horizontal gene transfer, and for the classification of unknown viral sequences, for example in an NGS setting where existing state-of-the-art methods for classification of reads, such as Kraken2[47] and Centrifuge[48], fail.

We finally suggest that a virus' virulence and gene expression pattern may be altered by modifying its genomic signature without altering its proteome. This opens a range of possibilities in the field of synthetic biology, such as for constructing attenuated viral vaccines, or for optimizing viral vectors used for gene therapy by changing the signatures of incorporated genes to fit that of the vector.

### Methods
#### Virus sequences

The dataset of eukaryotic viruses was based on the ICTV Master Species List (2018b.v2). The novel SARS-CoV-2 and two related sequences were later

added to the dataset. All available sequences from the ICTV Master Species List were downloaded from NCBI, and incomplete and partial sequences were excluded. Segmented genomes were treated as one separate entity per segment. This resulted in 4,273 sequences in the final dataset, representing 2768 unique taxonomic species (Supplementary Fig. 4, Supplementary Table 3). Low complexity regions, like tandem repeats, were removed from all original viral sequences using DustMasker[49] prior to all analyses, except for the sliding window analysis.

## Host sequences

We identified viral hosts using VirusHostDB[50], resulting in host assignments for 2519 out of 2768 viruses. We obtained the coding sequences of each host from NCBI. To create a single signature for every genome, we concatenated the coding sequences of each host with an interspersed $N$ character to avoid overlapping k-mers. The host dataset included 2399 eukaryotic genomes: 1267 fungi, 873 animals and insects, 234 plants, and 25 protists (Supplementary Table 4).

To eliminate the impact of endogenous viruses on hosts' genomic signature computation, we designed a bioinformatics protocol that excises the integrated elements from the hosts. Focusing on the insect viruses *Polydnaviridae*, we identified putative inserted regions by mapping host coding sequences on the virus genome with minimap2[51] (with the parameters "-cax map-ont –eqx –Y"). We excluded matching sequences when constructing the VLMCs.

## Genomic signatures

In this study, we use VLMC[17,18,52] and adopted a method, which was pioneered by Dalevi et al[6]. to model genomic signatures. These genomic signatures are an extension of the original concept, which was limited to dinucleotide frequencies[1]. Here, the Markov chain is a model which captures the relative frequency of individual nucleotides in a sequence, dependent on the k previous nucleotides, referred to as the context. Using chains of variable lengths, the VLMC instead captures frequencies of different oligonucleotides up to a length $k$ (k-mers). The algorithm from ref. 17 was used to compute the models. Specifically, the VLMC extends the Markov chain, which is a model that captures the likelihood of observing each nucleotide after seeing the previous $k$ nucleotides in a sequence. VLMCs allow for flexibility in context length, which can be longer for increased model specificity or shorter to simplify the model. Here, three parameters are used to control the size of the VLMC: the max depth which determines the largest number of previous nucleotides to consider, the min count that determines how frequent a context must be to be included, and the Kullback-Leibler threshold that determines which contexts are included in the VLMC. We aimed to have a context size that would encompass as many important genomic features as possible, while maintaining computability and accuracy of the models. We thus set the maximum depth parameter to 6, allowing the model to capture all nucleotide features up to a length of 6 in a genome, such as GC content, di-nucleotide motifs, codons, and codon pairs. We similarly selected a Kullback-Leibler threshold of 3.9075 (also the default value from ref. 6) and the min count was set to 2 to get enough support for the probabilities in the model. We further verified these parameter settings with the Bayesian information criterion[53], which is a method for selecting model complexity to avoid overfitting the VLMC to the profile.

To determine the presence and degree of specific genomic signatures in viruses, we split each viral sequence into two parts: query and profile, and VLMCs were calculated for the profile part. For each query, we computed the likelihood of observing it based on the VLMC profiles using the negative log-likelihood as a similarity measure, which was then normalized by sequence length. We classified a signature as species-, genus-, or family-specific based on the most similar profile's taxonomic group in relation to the query. We adjusted for different group sizes (see "Statistical analysis"). To ensure dissimilarity between homologous sequences, the query used for similarity measurements must not be part of the sequences used to compute VLMCs. We found that the profile length has a greater impact on results than the query length, and the query must be long enough to distinguish similar signatures. We empirically determined that a 30:70 (query:profile) ratio results in a good trade-off between these considerations.

## Tree-based clustering

To visualize genomic signature similarities and differences among viruses, we built an unrooted neighbor-joining tree[54]. The tree was based on a distance matrix containing the pairwise similarities between each *query* and *profile*, as previously calculated. Since the calculated distance d between two viruses A and B may differ slightly depending on which virus is query and which is profile, each distance D in the matrix was averaged as D(A,B) = ( d(A,B) + d(B,A))/2, and resulting distances were normalized by rescaling to a range from 0 to 1. We used Biopython[55] for the tree construction and ETE 3[56] for visualization.

## Sliding window

To conduct a comprehensive exploration of the conservation of genomic signatures throughout complete viral genomes, we devised a sliding window approach. To enhance sensitivity, we initially generated genomic profiles for all full viral sequences. Subsequently, specific regions (windows) within each query sequence were methodically examined as follows: An initial window, with a size of $n = 50$, was defined as the initial n nucleotides of the query sequence's genome. This window was then extracted from the query sequence, and a new profile was generated based on the remaining portion of the genome, which replaced the original profile derived from the complete genome in the profile dataset. The analysis was then carried out as previously described, comparing the genomic signature of the windowed sequence with all profiles to identify potential species, genus, or family specificity in the signature. By removing the windowed sequence from the profile genome of the same species, we mitigated homology-based biases that could lead to false positive species-specific matches.

The window was then shifted to the next position, which was a distance of 20% of the window size away from the starting point, and the analysis was repeated with the new window removed from the complete genome to generate an updated profile. This process was iterated until the end of the genome had been reached. Subsequently, the entire analysis was restarted from the beginning, utilizing a window that was 50 nucleotides larger. This process was repeated with windows increasing in size by 50 nucleotides for each iteration, until the window either reached a size of 10,000 nucleotides or one-third of the total genome length. For each window, the match between the window and the profile was plotted as a function of window size and color-coded based on specificity. At each position, the least accurate match from the overlap was selected to minimize false positives.

We randomly selected viruses that exhibited a species, genus, or family-specific signature from among viruses with genomes larger than 20,000 nucleotides. Specifically, we identified two viruses with family-specific signatures (*Duck atadenovirus A, Tupaiid betaherpesvirus 1*), two with genus-specific signatures (*Bat mastadenovirus B, Chroistoneura rosaceana nucleopolyhedrovirus*), and two with species-specific signatures (*Chelonus inanitus bracovirus, Cydia pomonella granulovirus*). (See Fig. 4a1–f1). Additionally, we plotted the fraction of windows exhibiting genomic signatures most similar to profiles of the same species, genus, or family to illustrate the length of sequence required to achieve a match (Fig. 4a2–c2).

To reduce computational resources, we also implemented a less resource-intensive version of the algorithm, where we randomly sampled 200 windows for each window size per virus sequence. This was applied to all viruses longer than 10,000 nucleotides to depict the specificity of signatures across various taxonomic ranks (see Supplementary Fig. 2).

## Statistical analysis

To obtain a comprehensive understanding, we employed a sampling methodology, which was reiterated 1000 times to gauge the significance of our findings. This approach randomly paired each query to a profile. Then, we computed the matches to the correct species, genus, and family for each virus, providing insight into the anticipated number of random matches, not considering the genomic signatures. We also conducted a Bonferroni-

corrected two-tailed *t*-test to compare the quantities of specific signatures in both the random analysis and the analyses based on genomic signatures. We considered adjusted *p* values less than 0.05 as statistically significant.

## Reporting summary

Further information on research design is available in the Nature Portfolio Reporting Summary linked to this article.

## Data availability

The data that support the findings in this study is available in GenBank. All accession IDs used are available as Supplementary Information for this paper. No new data was created in this study.

## Code availability

All custom code required to run our analysis is available at https://gitlab.com/genomic-signatures/genomic-signatures-in-viruses, https://doi.org/10.5281/zenodo.13907309.

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

## Acknowledgements

This project was partially funded by the Swedish Research Council through grant agreements no. 2015-05307 and 2018-05973, by FORMAS through grant agreement no. FR-2017/0009, by grants from the Swedish state under the agreement between the Swedish government and the county councils, the ALF-agreement (ALFGBG-971530, ALFGBG-971142, ALFGBG-932632, ALFGBG-725411), and The Swedish Society of Medicine (SLS-506371). Parts of the computations were enabled by resources provided by the Swedish National Infrastructure for Computing (SNIC) at NSC and C3SE.

## Author contributions

P.N. conceived of and supervised the study. All authors contributed to the design of the experiments, the analysis of the data and interpretation of the results. M.H. and J.G. implemented the methods under supervision by A.S. J.G. implemented the statistical analysis and host similarity analysis under supervision by A.S. M.H., J.G., and P.N. wrote the manuscript with input from all authors.

## Funding

## Competing interests

The authors declare no competing interests.
