## [Transparent Peer Review file · Communications Biology]

Evolution shapes and conserves genomic signatures in viruses

Corresponding Author: Professor Peter Norberg

Version 0:

Reviewer comments:

Reviewer #1

(Remarks to the Author)

The authors of this study analyzed genomic signatures in a diverse array of viruses that infect eukaryotes using advanced computational methods. Their goal was to understand the presence and specificity and conservation of unique genomic signatures within distinct taxonomic groups (species, genus, and family) and to assess the potential of these patterns in differentiating between various viruses.

The primary findings of this study show that genomic signatures in virus genomes exhibit high specificity, often discernible at the species level, and that these patterns frequently vary significantly even among species within the same family. The authors show that species-specificity is most pronounced among dsDNA viruses and those with larger genomes. The study further shows that viruses with larger viral genomes tend to exhibit more genomic signatures compared to their smaller counterparts. Given that genomic signatures in majority of the cases differ between viruses and their hosts, the authors deduce that the evolutionary selection pressures that govern the shaping of genomic signatures in viral genomes are distinct from those impacting the genomes of their hosts.

While the presented findings are somewhat interesting, it must be noted that this study does not provide many new insights, but rather affirms already known facts, or easily predictable outcomes. Furthermore, while the authors touch upon potential implications of their findings, these discussions remain largely speculative and somewhat far-fetched in nature. For instance, the idea that attenuated viral vaccines can be constructed by altering genomic signatures in virus genomes raises questions about the feasibility and practicality of such approaches.

Specific comments & questions

A genomic signature is a complex concept that encompasses various features, including the frequency and arrangement of di-nucleotides, codons and other sequence characteristics within a genome. In this study, the investigation of genomic signatures utilized an analysis of k-mer frequencies using variable-length Markov chain models. I believe that the process of identification of the genomic signatures should be better explained in the introduction or results section of the paper. Also, would it be possible to give an example of identified genomic signatures in different viruses to better illustrate the concept of genomic signatures?

While the methodology for determining species-specific genomic signatures is evident, there is an inadequate clarity regarding the process for identifying genus- and family-specific genomic signatures. It is essential to provide an explanation of how these broader taxonomic categories were determined and evaluated in the study.

The authors state that genomic signatures encompass features such as di-nucleotide, codon, or codon-pair preferences. It would be interesting to know whether di-nucleotide, codon or codon-pair preferences have the greatest influence on genomic signatures in different viruses.

The genomic signatures identified in this study had a maximum length of 6 nucleotides. Did the authors attempt to increase the maximum size of the k-mer to capture and model more complex signatures?

The analysis revealed a correlation between the length of viral genomes and the number of genomic signatures, with longer genomes showing a tendency to have more signatures compared to shorter genomes. Could you elaborate on the specific findings and potential implications of this correlation?

Viruses with larger genomes, such as herpesviruses, poxviruses, or iridoviruses, are known to harbor numerous repetitive sequences. These in turn can influence the number and complexity of genomic signatures in these viruses. Before

performing the genomic signature analysis, the authors used the DustMasker software to remove regions of low complexity from the viral sequences. It is advisable to explicitly mention this pre-processing step also in the results section. To what extent was the removal of the repetitive sequence successful, and were there any unexpected patterns or variations observed related to repetitive sequences?

Given that viruses with longer genomes often contain genes resulting from duplication events, did your study explore whether gene duplication played a role in the detection of genomic signatures?

The authors of this study identify genomic signatures within entire viral sequences. However, it is important to note that they exclusively utilize the protein-coding sequences of the virus host for genomic signature identification. Despite the generally high coding density in viruses, this methodological choice represents an approximation. Acknowledging this difference in approach is important. For improved comparability, it could be beneficial if the authors also analyzed only the protein-coding sequences of the viruses in their attempt to identify similarity in genomic signatures between the viruses and their hosts.

The authors state that they examined three randomly selected viruses with genomes ranging from 20,000 nt to 49,999 nt, and three viruses with genomes exceeding 50,000 nt to investigate intra-genomic variation in signature conservation (lines 189-191). However, it is not immediately evident which specific viruses were analyzed and what their genome sizes were. Furthermore, the results of this analysis on the six example viruses are presented in both the main figure (signatures in three viruses with genomes > 50 kb) and the supplementary figure (depicting signatures in three other viruses). This fragmentation of the data unnecessarily complicates the inspection and comprehension of the presented information.

Bracovirus genomes are typically substantial in size, often surpassing 100,000 nucleotides. It is not immediately evident that the authors treated each of the proviral segments of the *Chelonus inanitus* bracovirus as an independent entity. Perhaps this fact could be mentioned in the result section.

Polydnaviruses are characterized by a low coding density, a strong A/T bias and the presence of introns in many protein-coding genes. In addition, several of their genes show homology with genes found in insects or other eukaryotes. Some of these genes were recently acquired from wasps, while others were acquired by horizontal gene transfer from organisms outside the arthropod lineage. Hence, an alternative explanation for the observed similarity in genomic signatures between members of the Polydnaviridae family and their hosts may be that the detected host similarity does not originate from the integration of viral sequences into host genomes, or adaptation towards host's expression machinery, but rather from the incorporation of host sequences into the viral genome. Could the authors provide insights or comments on the validity of this alternative explanation?

Herpesviruses employ a variety of strategies to manipulate the host cell environment to their advantage. One such strategy involves the production of proteins known as host shutoff proteins. These proteins play a role in enhancing the expression of viral genes by downregulating host cell gene expression. Could it be that the vastly different genomic signatures observed in herpesviruses are a reflection of their adaptability and ability to thrive almost independently of the cellular environment?

Minor issues

Figure 2a

The formatting of the x-axis labeling should be corrected.

Legend of Figure 3

The legend contains a typo. Not order-specific, but genus-specific signatures were analyzed.

Extended Data Figure 2

Not blue, but dark green was used to depict windows with genus-specific genomic signatures.

Line 191

The results obtained for *Choristoneura rosaceana* nucleopolyhedrovirus are shown in Fig. 3b1, not Fig. 3c1.

Reviewer #2

(Remarks to the Author)

I have uploaded the review in a pdf format. You can find it in Review Attachments.

Editor comment: For a complete record, the content of the pdf is also placed here, for the formatted version see the attached pdf.

This paper aims to identify the genomic signature of viral species and evaluate its level of specificity. Authors characterize

genomic signatures using k-word frequencies, with $1 \leq k \leq 6$, which includes the genome composition of nucleotides, dinucleotides, codons and codon pairs, and classify each viral sequence as species-, genus-, or family-specific. They find that the majority of viral sequences are species-specific, with increasing specificity with increasing genome size. At the family level, however, they find some conflicting results, where some families show significant variations. They also evaluate if the signatures are related to each other according to the viral-host co-evolution, and find that signatures of viruses and those of their hosts show relevant dissimilarities. However, a qualitative comparison shows that viral signatures may be related due to common viral-host dependencies. Authors conclude that different selection pressures contribute to the formation of robust genomic signatures.

The paper is well written and of high scientific quality. The content is clear and the methodology is appropriate. Furthermore, the results provide new insights in the field of virology and comparative biology, which may be relevant for the implementation of new methodologies in viral classification. I only have a few minor comments, which I consider should be reviewed.

1. I suggest to emphasize that genome signatures based on word frequencies are at the core of alignment-free methods in comparative biology. It is well known that related organisms share similar word abundances, so genomic signatures have been applied in numerous studies to identify evolutionary relationships and classify metagenomic data. Relevant studies that can be cited are [1, 2].
2. Some studies compare genomic sequences by computing pairwise distances and applying a clustering method, such as the principal component analysis (PCA). I would find interesting if authors can explain the advantages of their methodology.
3. Numerous studies compare DNA sequences using a fixed word length. There is not a priori knowledge of the optimum length for sequence comparison, such it depends on species word usage, the fragment size to analyze, and the method for word count. However, it has been observed that longer words are required to discriminate between phylogenetically close species. If specificity increases when increasing word sizes, discrimination at different taxonomical levels may occur by varying the word size. Studies that analyze the effect of the word length include [3, 4]. In this study, authors compare sequences using variable order Markov models of genomic signatures with word sizes up to $k = 6$, and use different statistical methods to determine the taxonomical specificity. The methods are appropriate and take into account the dependency on neighbour composition. However, it would be interesting if authors discuss the effect of the word length in taxonomical classifications.
4. Due to a lack of molecular tools, viruses have been commonly classified based on morphological traits, such as characteristics in their life cycle or virus-host dependencies. However, the use of alignment-free methods has allowed important advances in the taxonomical classification of metagenomic data and genomes with mosaic patterns. Specifically, these methods are very useful for viral classification. Nowadays, I would always recommend to take into account both morphological and genomic traits when comparing viral genomic sequences. For example, the viral condition of being lytic or lysogenic determines to a great extent the genomic distance to their hosts [5, 6]. It would be interesting if authors can evaluate or discuss the effect of the life cycle in the virus-host pairwise distances.
5. It has been documented that viruses are related to each other according to common host dependencies [7]. In particular, the host range influences the specificity of viral genomic signatures and the distances to their hosts. Viruses that infect a wide range of different hosts lose their degree of specificity. It would be also interesting if authors can discuss the influence of the host range in the genomic signatures.
6. Authors only use the coding regions of the hosts and exclude all endogenous viral sequences. However, these restrictions may mask some events of horizontal gene transfer, which plays a key role in genome evolution and contributes to the genomic signature. I suggest to add a comment about the implications of these conditions.
7. In line 128. Describe what "simulation approach" means in this context.
8. In line 264. I suggest to change "distinct signatures" by other expression, such as "specific signatures". "Distinct" here is arbitrary.
9. In line 335. Describe better what does it mean "...more diverged in sequence.".
10. In line 415 we can read "We finally suggest that a virus' phenotype may be altered by modifying its genomic signature without altering its proteome." Define "virus' phenotype". I think that "virus' phenotype" refers here to gene expression patterns. However, the term is not clear in this context. I suggest to describe it better.
11. In line 577. The general reader is not familiar with the terminology related to Markov processes. Specify that the term "context length" refers to the number of nucleotides to consider in the Markov chain.
12. Increase image resolution in Extended Data Figure 1-3.
13. Increase the figure legend in Extended Data Figure 4. 2
14. Rewrite columns of Supplementary Data 3-4 with all information.

References

- [1] Rebeca de la Fuente, Wladimiro Díaz-Villanueva, Vicente Arnau, and Andrés Moya. Genomic signature in evolutionary biology: A review. *Biology*, 12(2), 2023.
- [2] S Karlin and I Ladunga. Comparisons of eukaryotic genomic sequences. *Proceedings of the National Academy of Sciences*, 91(26):12832–12836, 1994.
- [3] P. Deschavanne, A. Giron, J. Vilain, C. Dufraigne, and B. Fertil. Genomic signature is preserved in short dna fragments. In *Proceedings IEEE International Symposium on Bio-Informatics and Biomedical Engineering*, pages 161–167, 2000.
- [4] Charles Chapus, Christine Dufraigne, Scott Edwards, Alain Giron, Bernard Fertil, and Patrick Deschavanne. Exploration of phylogenetic data using a global sequence analysis method. *BMC Evolutionary Biology*, 5(1):1–18, 2005.
- [5] Patrick Deschavanne, Michael S DuBow, and Christophe Regnard. The use of genomic signature distance between bacteriophages and their hosts displays evolutionary relationships and phage growth cycle determination. *Virology Journal*, 7(163):1–12, 2010.
- [6] Vicente Arnau, Wladimiro Díaz-Villanueva, Jorge Mifsut Benet, Paula Villasante, Beatriz Bea mud, Paula Mompó, Rafael Sanjuan, Fernando González-Candelas, Pilar Domingo-Calap, and Mária Džunková. Inference of the life cycle of environmental phages from genomic signature distances to their hosts. *Viruses*, 15(5), 2023.

[7] Travis N Mavrich and Graham F Hatfull. Bacteriophage evolution differs by host, lifestyle and genome. *Nature Microbiology*, 2(9):1–9, 2017.

Author Rebuttal letter:

Reply to referees' comments

We would like to express our sincere gratitude to the two referees that have taken their time to read our manuscript and for giving us valuable suggestions for improvement. Each comment has been addressed and our comments are in red text below. Edited sections in the manuscript are marked with vertical red lines in the left margin. Please note that we have added two more figures (Fig. 3 and Extended Data Figure 1), and that previous Extended Data Fig. 2 was merged with Fig. 4 (previous Fig. 3) based on referee suggestions.

Reviewers' comments:

Reviewer #1 (Remarks to the Author):

The authors of this study analyzed genomic signatures in a diverse array of viruses that infect eukaryotes using advanced computational methods. Their goal was to understand the presence and specificity and conservation of unique genomic signatures within distinct taxonomic groups (species, genus, and family) and to assess the potential of these patterns in differentiating between various viruses.

The primary findings of this study show that genomic signatures in virus genomes exhibit high specificity, often discernible at the species level, and that these patterns frequently vary significantly even among species within the same family. The authors show that species-specificity is most pronounced among dsDNA viruses and those with larger genomes. The study further shows that viruses with larger viral genomes tend to exhibit more genomic signatures compared to their smaller counterparts. Given that genomic signatures in majority of the cases differ between viruses and their hosts, the authors deduce that the evolutionary selection pressures that govern the shaping of genomic signatures in viral genomes are distinct from those impacting the genomes of their hosts.

While the presented findings are somewhat interesting, it must be noted that this study does not provide many new insights, but rather affirms already known facts, or easily predictable outcomes. Furthermore, while the authors touch upon potential implications of their findings, these discussions remain largely speculative and somewhat far-fetched in nature. For instance, the idea that attenuated viral vaccines can be constructed by altering genomic signatures in virus genomes raises questions about the feasibility and practicality of such approaches.

- We agree that a discussion of results should generally not be too speculative and far-fetched in nature. However, although some suggested implications might seem somehow speculative at a first glance, we do not see them as too far-fetched to be mentioned in the discussion. For example, attenuating viruses by altering parts of their genomic signatures (codon-pair bias and/or di-nucleotide motifs) has already been performed successfully by several groups, as mentioned in the introduction of the manuscript. It is also highly likely that viral genomes are subject to selection pressures that favor and restrict certain mutations, which we believe may have implications in, for example, evolutionary studies as we discuss. Since we present these implications merely as suggestions, we would like to keep this part of the discussion in its present form.

Specific comments & questions

A genomic signature is a complex concept that encompasses various features, including the frequency and arrangement of di-nucleotides, codons and other sequence characteristics within a genome. In this study, the investigation of genomic signatures utilized an analysis of k-mer frequencies using variable-length Markov chain models. I believe that the process of identification of the genomic signatures should be better explained in the introduction or results section of the paper. Also, would it be possible to give an example of identified genomic signatures in different viruses to better illustrate the concept of genomic signatures?

- We are thankful for this comment and agree that the concept can be better explained. We have therefore expanded the explanation of genomic signatures in the first paragraph of the Results section. As suggested, we have also added a figure (Extended Data Figure 1) illustrating how Markov models are built, and that depicts the signatures of two different viruses represented by their different k-mers.

While the methodology for determining species-specific genomic signatures is evident, there is an inadequate clarity regarding the process for identifying genus- and family-specific genomic signatures. It is essential to provide an explanation of how these broader taxonomic categories were determined and evaluated in the study.

- We agree that this could be better explained and have extended the text under “Genomic signatures in viral genomes” in the results section for clarification. In short, most viral species that we stated present genus- or family- specific genomic signatures in fact have conserved signatures within their genomes, but not enough different from the signature in at least one other species in the same genus or family to distinguish them. We hope that this will now be clearer in the manuscript.

The authors state that genomic signatures encompass features such as di-nucleotide, codon, or codon-pair preferences. It would be interesting to know whether di-nucleotide, codon or codon-pair preferences have the greatest influence on genomic signatures in different viruses.

- This is intriguing question that we have thought about a lot ourselves. It is, however, not trivial to point out which parts that have the most influence on the signature or has the most impact on various biological properties. As mentioned in the introduction, it has been debated whether attenuation of a virus is caused by de-optimization of codon-pair-bias, or di-nucleotide motifs, since they are connected, i.e. you cannot change one without affecting the other. In a similar manner, the codon usage is also linked to the GC-content. Since the signatures, including GC-content, vary between viruses, it is also reasonable to believe that different characteristics are of different importance for different viruses. Although we feel that it is beyond the scope of the present paper, an in-depth analysis of the signatures of certain viral species, and comparisons between species, may reveal important information on various selection pressures and functions. With that said, we have, like several previous studies, noticed that the GC-content has a significant overall role in the genomic signatures. We have extended the discussion to address this important question, and we hope that the added figure illustrating the different signatures in two viruses may further illuminate this topic.

The genomic signatures identified in this study had a maximum length of 6 nucleotides. Did the authors attempt to increase the maximum size of the k-mer to capture and model more complex signatures?

- We have included a re-analysis using different max-lengths of the k-mers (one to seven). Our results are presented in the results section with an additional figure (Fig. 3) and the topic is addressed in the discussion.

The analysis revealed a correlation between the length of viral genomes and the number of genomic signatures, with longer genomes showing a tendency to have more signatures compared to shorter genomes. Could you elaborate on the specific findings and potential implications of this correlation?

- We are not certain that we fully understand this comment. What we found in this study was that the predictive power to determine the correct species, genus, or family for a certain virus, based on its genomic signature, was stronger for species with large genomes compared to those with smaller genomes. This finding was valid even when we analyzed subregions of the genomes. We discuss possible explanations to these findings in the discussion and suggest that it is partly a methodological issue, but that there also seem to be differences when we analyze sub-regions of viruses with large genomes. Although we have no explicit explanation to why we found these results, we do discuss different possibilities.

Viruses with larger genomes, such as herpesviruses, poxviruses, or iridoviruses, are known to harbor numerous repetitive sequences. These in turn can influence the number and complexity of genomic signatures in these viruses. Before performing the genomic signature analysis, the authors used the DustMasker software to remove regions of low complexity from the viral sequences. It is advisable to explicitly mention this pre-processing step also in the results section. To what extent was the removal of the repetitive sequence successful, and were there any unexpected patterns or variations observed related to repetitive sequences?

- We have added information about DustMasker in the results section as requested.

- We have observed that long regions with many repeats can bias the results and that the results significantly improved after using DustMasker. The reason is simply that the relative frequency of every word of length ≤ 6 contained in the repeated blocks will be overrepresented in the signature due to the repeats, which is why we pre-processed the sequences with DustMasker prior to further analyses. An exception is the sliding window analysis which were performed on non-trimmed sequences to highlight the changes in signature in repeat regions (consequently usually red regions in the figures).

- We cannot state for sure to what extent the removal of repetitive sequences was successful since this would require manual inspection of each sequence, which would be time-wise impractical. Moreover, even with such manual inspection, we would have to set certain definitions of regions to remove, which is why we, for simplicity, and to use established standards, used the definitions formulated for DustMasker.

- We would like to stress, however, that a repeat region present exclusively in either one of the query or profile sequences would infer a bias that would likely give a miss-match between the two. As such, the existence of repeat regions would likely give lower specificity than obtained in the present study, implying that a more rigorous manual removal of repeat regions would rather improve the specificity of our results, than to lower it.

Given that viruses with longer genomes often contain genes resulting from duplication events, did your study explore whether gene duplication played a role in the detection of genomic signatures?

- This is an interesting question that we have considered, but we didn't do any in-depth analysis on this matter in the present study. Our sliding window analysis suggests, however, that the genomic signatures are generally conserved throughout the genomes, implying that gene duplication has little or no effect on our results. Multiple copies of a gene, should they exist, would change the overall k-mer frequencies, which would likely yield worse results when analyzing regions not harboring copies of the repeated genes. We cannot, however, exclude completely the possibility that a slight bias might occur in rare cases where paralog genes (that have not diverged significantly on the DNA or RNA sequence level) are present both in the query and the profile sequences. We have added a comment about this possible limitation in the discussion.

The authors of this study identify genomic signatures within entire viral sequences. However, it is important to note that they exclusively utilize the protein-coding sequences of the virus host for genomic signature identification. Despite the generally high coding density in viruses, this methodological choice represents an approximation. Acknowledging this difference in approach is important. For improved comparability, it could be beneficial if the authors also analyzed only the protein-coding sequences of the viruses in their attempt to identify similarity in genomic signatures between the viruses and their hosts.

- We acknowledge this to be a valid point, although many viruses have high coding density (as stated in the comment above) and overlapping reading frames that also go in different directions, which should limit the bias. Furthermore, the sliding window analysis suggests that the signature within a coding region is not vastly different from the signature of the complete genome. Nevertheless, we have added an acknowledgement of this approximation in the new section "Possible limitations" in the discussion.

The authors state that they examined three randomly selected viruses with genomes ranging from 20,000 nt to 49,999 nt, and three viruses with genomes exceeding 50,000 nt to investigate intra-genomic variation in signature conservation (lines 189-191). However, it is not immediately evident which specific viruses were analyzed and what their genome sizes were. Furthermore, the results of this analysis on the six example viruses are presented in both the main figure (signatures in three viruses with genomes > 50 kb) and the supplementary figure (depicting signatures in three other viruses). This fragmentation of the data unnecessarily complicates the inspection and comprehension of the presented information.

- This has been clarified and the two figures have been merged into one to make it less complicated to inspect (Fig. 4).

Bracovirus genomes are typically substantial in size, often surpassing 100,000 nucleotides. It is not immediately evident that the authors treated each of the proviral segments of the *Chelonus inanitus* bracovirus as an independent entity. Perhaps this fact could be mentioned in the result section.

- We state in the first paragraph of the result section that each segment was analyzed separately for segmented viruses, but we agree that it might not be immediately evident in each case. For further clarification regarding the *Chelonus inanitus* bracovirus, we have added information about this in the result section.

Polydnaviruses are characterized by a low coding density, a strong A/T bias and the presence of introns in many protein-coding genes. In addition, several of their genes show homology with genes found in insects or other eukaryotes. Some of these genes were recently acquired from wasps, while others were acquired by horizontal gene transfer from organisms outside the arthropod lineage. Hence, an alternative explanation for the observed similarity in genomic signatures between members of the Polydnaviridae family and their hosts may be that the detected host similarity does not originate from the integration of viral sequences into host genomes, or adaptation towards host's expression machinery, but rather from the incorporation of host sequences into the viral genome. Could the authors provide insights or comments on the validity of this alternative explanation?

- This is a very interesting suggestion, and although we have not performed an in-depth analysis in the present study, we have added an acknowledgment of this alternative explanation in the discussion.

Herpesviruses employ a variety of strategies to manipulate the host cell environment to their advantage. One such strategy involves the production of proteins known as host shutoff proteins. These proteins play a role in enhancing the expression of viral genes by downregulating host cell gene expression. Could it be that the vastly different genomic signatures observed in herpesviruses are a reflection of their adaptability and ability to thrive almost independently of the cellular environment?

- This is an interesting point. Although we believe that the genomic signatures in herpesviruses, and other viral families, are shaped by a multitude of various selection pressures, we agree that host shutoff proteins indeed may induce a host cell independence that can facilitate a divergence of these. We have added a comment about this in

the discussion.

Minor issues

Figure 2a

The formatting of the x-axis labeling should be corrected.

- This has been corrected.

Legend of Figure 3

The legend contains a typo. Not order-specific, but genus-specific signatures were analyzed.

- This has been corrected.

Extended Data Figure 2

Not blue, but dark green was used to depict windows with genus-specific genomic signatures.

- This has been corrected and the figure have been merged with the other sliding window figure as suggested above (Fig. 4).

Line 191

The results obtained for *Choristoneura rosaceana* nucleopolyhedrovirus are shown in Fig. 3b1, not Fig. 3c1.

- This has been corrected.

Reviewer #2 (Remarks to the Author):

This paper aims to identify the genomic signature of viral species and evaluate its level of specificity. Authors characterize genomic signatures using k-word frequencies, with $1 \leq k \leq 6$, which includes the genome composition of nucleotides, dinucleotides, codons and codon pairs, and classify each viral sequence as species-, genus-, or family-specific. They find that the majority of viral sequences are species-specific, with increasing specificity with increasing genome size. At the family level, however, they find some conflicting results, where some families show significant variations. They also evaluate if the signatures are related to each other according to the viral-host co-evolution, and find that signatures of viruses and those of their hosts show relevant dissimilarities. However, a qualitative comparison show that viral signatures may be related due to common viral-host dependencies. Authors conclude that different selection pressures contribute to the formation of robust genomic signatures. The paper is well written and of high scientific quality. The content is clear and the methodology is appropriate. Furthermore, the results provide new insights in the field of virology and comparative biology, which may be relevant for the implementation of new methodologies in viral classification. I only have a few minor comments, which I consider should be reviewed.

1. I suggest to emphasize that genome signatures based on word frequencies are at the core of alignment-free methods in comparative biology. It is well known that related organisms share similar word abundances, so genomic signatures have been applied in numerous studies to identify evolutionary relationships and classify metagenomic data. Relevant studies that can be cited are [1, 2].

- We have included these relevant references in the first paragraph of the introduction.

2. Some studies compare genomic sequences by computing pairwise distances and applying a clustering method, such as the principal component analysis (PCA). I would find interesting if authors can explain the advantages of their methodology.

- We do pairwise comparison that we represent in the neighbor joining (NJ) tree in the manuscript. We acknowledge that there are alternative methods to present similarities of genomic signatures, such as PCA, but decided to present our pairwise distances using NJ since it is an established method to graphically present pairwise distances, although traditionally used for phylogenetic purposes. We have not, however, performed any explicit comparative analysis between or method and others, but a comparison could be interested in future studies with a more methodological focus.

3. Numerous studies compare DNA sequences using a fixed word length. There is not a priori knowledge of the optimum length for sequence comparison, such it depends on species word usage, the fragment size to analyze, and the method for word count. However, it has been observed that longer words are required to discriminate between phylogenetically close species. 1 If specificity increases when increasing word sizes, discrimination at

different taxonomical levels may occur by varying the word size. Studies that analyze the effect of the word length include [3, 4]. In this study, authors compare sequences using variable order markov models of genomic signatures with word sizes up to $k = 6$, and use different statistical methods to determine the taxonomical specificity. The methods are appropriate and take into account the dependency on neighbour composition. However, it would be interesting if authors discuss the effect of the word length in taxonomical classifications.

- The impact of word size is an interesting topic and indeed depends on various variables, such as species, genome size, sequence length under analysis, etc. We have performed an additional analysis where we compared the results using different word max-sizes (up to seven nucleotides) applied on viral genomes of different sizes (Fig. 3) and have extended the discussion to address this topic. The mentioned studies have also been cited.

4. Due to a lack of molecular tools, viruses have been commonly classified based on morphological traits, such as characteristics in their life cycle or virus-host dependencies. However, the use of alignment-free methods has allowed important advances in the taxonomical classification of metagenomic data and genomes with mosaic patterns. Specifically, these methods are very useful for viral classification. Nowadays, I would always recommend to take into account both morphological and genomic traits when comparing viral genomic sequences. For example, the viral condition of being lytic or lisogenic determines to a great extent the genomic distance to their hosts [5, 6]. It would be interesting if authors can evaluate or discuss the effect of the life cycle in the virus-host pairwise distances.

- This is a very interesting topic, and we believe that an increased knowledge about the different biological traits underlying the various selection pressures affecting the genomic signatures of different viruses would be highly appreciated in many fields. A complete comparison and evaluation of the effect of various biological traits, such as the life cycle in the virus-host pairwise distances, for all viruses included in this study would, however, demand a tremendous work that we consider to be more suitable to address in future studies. Although bacteriophages, which the suggested studies focused on, differ in many ways from eucaryotic viruses that we analyzed here, we are thankful for the suggestion and have added a section in the discussion addressing this interesting topic.

5. It has been documented that viruses are related to each other according to common host dependencies [7]. In particular, the host range influences the specificity of viral genomic signatures and the distances to their hosts. Viruses that infect a wide range of different hosts lose their degree of specificity. It would be also interesting if authors can discuss the influence of the host range in the genomic signatures.

- This is a very interesting question, and while several studies on bacteriophages have presented data supporting viral-host adaptation and correlation with host range, we were not able to find convincing patterns of such correlation in the present study on eucaryotic viruses. We have extended the discussion to address this interesting topic under "Variations within and between families".

6. Authors only use the coding regions of the hosts and exclude all endogenous viral sequences. However, these restrictions may mask some events of horizontal gene transfer, which plays a key role in genome evolution and contributes to the genomic signature. I suggest to add a comment about the implications of these conditions.

- We agree about the comment on only using coding regions of the hosts, as was also commented on by reviewer 1, and has added information about this (please see comment to reviewer 1 above).

- Regarding the comment about masking events of horizontal gene transfer by excluding endogenous viral sequences, this was a deliberate choice to limit bias. We agree that horizontal gene transfer plays a significant role in evolution, but, if these incorporated sequences have not yet mutated enough to adapt to the new genome, inclusion of these will also affect the overall signature in an un-wanted way (hence the exclusion). If, however, the endogenous virus sequence infers functions to the new genome that will change the overall selection pressure that shapes the genomic signature of the host genome and the genomes of infecting viruses, we agree that it might be interesting to analyze these separately. Although we believe that such analysis is outside the scope of the current study, we have acknowledged this possibility in the new "Possible limitations" section in the end of the discussion.

7. In line 128. Describe what "simulation approach" means in this context.

- A clarification has been added.

8. In line 264. I suggest to change "distinct signatures" by other expression, such as "specific signatures". "Distinct" here is arbitrary.

- Ok

9. In line 335. Describe better what does it mean "...more diverged in sequence.".

- This has been changed to "more distantly related"

10. In line 415 we can read "We finally suggest that a virus' phenotype may be altered by modifying its genomic signature without altering its proteome." define "virus' phenotype". I think that "virus' phenotype" refers here to gene expression patterns. However, the term is not clear in this context. I suggest to describe it better.

- This has been changed to "virulence and gene expression pattern"

11. In line 577. The general reader is not familiar with the terminology related to Markov processes. Specify that the term "context length" refers to the number of nucleotides to consider in the Markov chain.

- A clarification has been added.

12. Increase image resolution in Extended Data Figure 1-3.

- Extended Data Figure 2 has been merged and moved to main manuscript, as suggested by reviewer 1 (Fig. 4). Images of higher resolution will be provided prior to publication.

13. Increase the figure legend in Extended Data Figure 4. 2

- The figure legend has been increased.

14. Rewrite columns of Supplementary Data 3-4 with all information.

- It is, unfortunately, not entirely clear to us what information is missing. The accession numbers are listed, giving the reader the possibility to achieve any available data. However, including all information would result in extremely large tables, which would not be very practical. If we have misunderstood the comment and if there are some specific data missing, we would be happy to add it.

References

[1] Rebeca de la Fuente, Wladimiro Díaz-Villanueva, Vicente Arnau, and Andrés Moya. Genomic signature in evolutionary biology: A review. *Biology*, 12(2), 2023.

[2] S Karlin and I Ladunga. Comparisons of eukaryotic genomic sequences. *Proceedings of the National Academy of Sciences*, 91(26):12832–12836, 1994.

[3] P. Deschavanne, A. Giron, J. Vilain, C. Dufraigne, and B. Fertil. Genomic signature is preserved in short dna fragments. In *Proceedings IEEE International Symposium on Bio-Informatics and Biomedical Engineering*, pages 161–167, 2000.

[4] Charles Chapus, Christine Dufraigne, Scott Edwards, Alain Giron, Bernard Fertil, and Patrick Deschavanne. Exploration of phylogenetic data using a global sequence analysis method. *BMC Evolutionary Biology*, 5(1):1–18, 2005.

[5] Patrick Deschavanne, Michael S DuBow, and Christophe Regeard. The use of genomic signature distance between bacteriophages and their hosts displays evolutionary relationships and phage growth cycle determination. *Virology Journal*, 7(163):1–12, 2010.

[6] Vicente Arnau, Wladimiro Díaz-Villanueva, Jorge Mifsut Benet, Paula Villasante, Beatriz Bea mud, Paula Mompó, Rafael Sanjuan, Fernando González-Candelas, Pilar Domingo-Calap, and Mária Džunková. Inference of the life cycle of environmental phages from genomic signature distances to their hosts. *Viruses*, 15(5), 2023.

[7] Travis N Mavrich and Graham F Hatfull. Bacteriophage evolution differs by host, lifestyle and genome. *Nature Microbiology*, 2(9):1–9, 2017.

Version 1:

Reviewer comments:

Reviewer #1

(Remarks to the Author)

The authors have satisfactorily addressed all of my questions and comments. They have also improved the discussion section by discussing the limitations of the study. I have only minor comments on the study.

As recommended, the authors provided an example of genomic signatures determined for two different viruses using a variable-length Markov chain model. Since this figure effectively illustrates the essence of genomic signatures –

identification of significant and representative k-mer patterns in genomic data – I suggest moving it to the main text.

Figure 3 shows how the maximum k-mer length impacts the accuracy of correctly identifying the species, genus, or family from which the k-mers were derived. While the results for each category are displayed in separate columns, the small font used for labels makes it difficult to distinguish between them. To improve clarity, I recommend labeling each column more clearly to indicate its corresponding category.

Finally, virus family names such as “Hepadna, Nudi, and Asco” etc. (line 300) should be written out in full.

Reviewer #2

(Remarks to the Author)

The authors have satisfactorily addressed all my questions and concerns.
